# Assessment of Sequential Yeast Inoculation for Blackcurrant Wine Fermentation

**Zhuoyu Wang [1,\*], Andrej Svyantek [1], Zachariah Miller [1] and Aude A. Watrelot [2]**

[1] Western Agriculture Research Center, Montana State University, Corvallis, MT 59828, USA; # andrej.svyantek@montana.edu (A.S.); zachariah.miller@montana.edu (Z.M.)

[2] Department of Food Science and Human Nutrition, Iowa State University, Ames, IA 50011, USA; # watrelot@iastate.edu

\* Correspondence: zhuoyu.wang@montana.edu

**Abstract:** Blackcurrant is well known for its health benefits, but its wine products are understudied. In this research, studies were conducted after non-*Saccharomyces* and *Saccharomyces* yeast strain inoculation in less than 20% (*w/v*) fruit must for blackcurrant fermentation. Three inoculations were carried out on blackcurrant musts, as follows: (1) sequential inoculation with *Torulaspora delbrueckii* (strain Biodiva) followed by *Saccharomyces* EC1118 strain; (2) sequential inoculation with *Metschnikowia pulcherimma* (strain Flavia) followed by EC1118; (3) single-strain inoculation with EC1118 as the control treatment. None of these treatments did alter sugar consumption dynamics. Biodiva inoculation had impacts on both color dynamic parameter changes and final wine color profiles compared to EC1118. The final wine compositions indicate that Biodiva treatment had a significant impact on wine pH and acidity, whereas EC1118 single-strain largely influenced wine ethanol and glycerol contents. Although the total antioxidant capabilities were close among the three produced wines, the monophenol profiles indicate that Biodiva enhanced the total anthocyanin and hydroxycinnamates content but reduced the total flavanol contents in the final wine. EC1118 and Flavia wines contained more total flavanols compared to Biodiva treatment. The nonflavonoid profiles indicate that there were no significant differences among the three treatments. Our findings provide useful information for the application of yeast strains in blackcurrant wine fermentation.

**Keywords:** blackcurrant wine; non-*Saccharomyces*; *Saccharomyces cerevisiae*

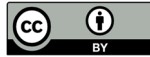

## 1. Introduction

Blackcurrants (*Ribes nigrum* L.) are mainly grown in Europe; they are used for processed food products such as juice, jams, and wines [1]. Blackcurrants have been well-known for their health-promoting and disease-preventing properties [2]. Many polyphenols contribute to the health and sensory attributes that define blackcurrants; in particular, the anthocyanins provide strong, vibrant colors and antioxidant activities. One of the key blackcurrant aromatic compounds is 4-mercapto-4-methyl-pentan-2-one (4MMP), which contributes to the "green", "mint", and "exotic fruits" descriptions [3]. Vitamin C (L-ascorbic acid) is another representative antioxidant component in blackcurrants [4].

Fruit wines are made from various fruits that are not grapes (*Vitis* spp.), and their markets are increasing due to the diversified needs of consumers. Fruit wines as value-added products also aid with imbalances between fruit production and marketing [5,6]. In cold climates where European grapevines (*V. vinifera* L.) cannot survive, cold hardy fruits, such as blackcurrants, expand the production options for wine makers and processors. Blackcurrant processed products, such as wines, retain compounds that may benefit human health, mainly because of their high amounts of total phenolics and an-

thocyanins [7]. Blackcurrant wines are reported as high in total antioxidant capacity (more than 1500 mg ascorbic acid equivalents/L) and total phenolic content (more than 1500 mg gallic acid equivalents/L) [8]. Beyond these key compounds, blackcurrant fruit and wines also contain vitamins, provitamins, minerals and phytosterols [9].

In the wine industry, one of many challenges is the creation and maintenance of high-quality wine, regardless of starting materials. Multiple food processing and fermentation techniques have been used in blackcurrant fermentation to alter the physico-chemical properties of the final wines. Previous work examining heat maceration, maceration with enzymatic treatments, and the combination of enzymes and different temperatures for their impact on blackcurrant wine chemical composition noted the impacts on total anthocyanin concentration [10]. Anthocyanins are the main contributors to wine color and can have a significant impact on wine's visible sensory evaluation. Color spectrum measurements are typically used to judge the wine's appearance without the bias of sensory evaluation [11]. Beyond maceration techniques, microbes have been employed, such as *Saccharomyces cerevisiae*, *Saccharomyces bayanus*, and *Torulaspora delbrueckii*; these species have been previously tested in the fermentation of two blackcurrant varieties [12]. The results indicate that microbial choices had significant effects on the final anthocyanin concentration and ethanol content. Finnish researchers have also focused on yeast strains' impact on chemical composition [13]. Beyond anthocyanins, the enzymatic treatment of blackcurrant pulp alters the extraction of polyphenol compounds and increases the antioxidative phenols in the blackcurrant wines [14,15].

For blackcurrant fermentation research with *Saccharomyces* and non-*Saccharomyces*, previous work used pasteurized blackcurrant pressed juice as the fermentation must starting material, with results mainly focused on the final wine volatile compounds [13]. The sequential inoculation of non-*Saccharomyces* and *Saccharomyces cerevisiae* expands potential wine styles [16]. The typical northern U.S.A. fruit wines are usually made from diluted fruit juice or water-ameliorated whole fruit due to the relatively high acidity in many fruits, such as *Prunus cerasus* (sour cherry), *Lonicera caerulea* (haskap), blackcurrants, and more. This study used diluted blackcurrant fruit juice for the fermentation starting material. However, based on the current knowledge gaps, there is a need to evaluate the effect of sequential inoculation on blackcurrant wine chemistry, as characterized through fermentation physiochemical and color dynamic changes as well as core antioxidant content and activity.

There are two non-*Saccharomyces* strains (*T. delbrueckii* and *Metschnikowia pulcherrima*) included in this study to compare with the *Saccharomyces* yeast strain's effects. One species, *T. delbrueckii*, has the capacity to ferment in monoculture [17]. Biodiva was chosen as a representative commercial strain of *T. delbrueckii*, which is reported to improve red and white wine aromas [18]. It can add some complexity to the wine aroma, and compounds such as esters, higher alcohols, and glycerol. It is usually co-inoculated or sequentially inoculated with other *Saccharomyces* yeast strains [19]. Introducing *T. delbrueckii* into fermentation can have positive effects, such as increasing the color intensity in the final wines [20]. It also has the capacity to produce other metabolites, such as glycerol, to improve wine complexity and mouth-feel [21]. Varying ratios of Biodiva and *S. cerevisiae* strain SafAle™ S-04 were associated with the variable potential in fermented products [22]. The other species, *Metschnikowia pulcherrima*, has an impact on wine aromas, such as terpenes and volatile thiols; for *M. pulcherrima*, Flavia is the commercialized pure culture strain that was selected for use. It has been reported to reduce the final wine alcohol content due to its glucose oxidase activities [23]. However, *M. pulcherrima* has low ethanol tolerance, which can alter the rate of fermentation [24]. *M. pulcherrima* has the potential to increase glycerol and aromatic compounds in wines. Meanwhile, *M. pulcherrima* treatment has been shown to impact organic acid concentration, antioxidant activity, and color stability in dragon fruit wines [25]. The chemical compositions of durian wines have also been impacted by inoculation with *M. pulcherrima* (Flavia), *T. delbrueckii* (Biodiva), *Pichia kluyveri* (FrootZen), and *Kluyveromyces thermotolerans* (Concerto) [26].

In this study, diluted blackcurrant materials were used for fermentation starting materials, which were ameliorated with water followed by sugar adjustment, to target typical alcohol ranges, and acid additions to achieve stable pH ranges. Through the assessment of carbohydrate consumption, pH, color parameters, total phenols, antioxidant content and activity changes, the uses of *T. delbrueckii* and *M. pulcherrima* as part of the sequential inoculation of blackcurrant wine fermentation dynamics and final wines were thoroughly assessed. This research provides valuable information on the application of yeast strain sequential inoculation for cold-hardy fruit wines.

## 2. Materials and Methods

### 2.1. Fermentation Procedures

"Stikine" blackcurrants were planted at the Western Agricultural Research Center of Montana State University, Corvallis, MT in 2015. Ripe blackcurrant fruits were harvested during August 2022 with berry soluble solid °Brix of 16.0–20.0 and pH of 2.76–3.19, and the fruit were stored in a freezer (−20 °C).

Before fermentation, frozen blackcurrants were thawed overnight in a walk-in cooler (Norlake, Hudson, WI, USA) with a temperature between 1 and 4 °C and humidity >90% (Figure 1). A Hobo logger (MX2301A, HOBO, MA, USA) was used to monitor the temperature and humidity.

The next day, the blackcurrants were processed by a centrifugal fruit juicer (Breville, Torrance, CA, USA), and the juice and pulps were homogenized afterward. Half a kilogram of the mixture was added to each little big mouth bubbler used as the fermenter (Northern Brewer, West Roseville, MN, USA). In total, 9 fermenters were included in this study for three treatments and three fermentation replicates of each treatment. Following the addition of blackcurrant materials into each fermenter, 0.5 L of drinking water (Signature Select, Ocala, FL, USA) was added to each fermenter and mixed thoroughly. Meanwhile, Scottzyme HC (0.35 mL/L, Scott Labs, Lubbock, TX, USA), Scottzyme PEC 5L (0.02 mL/L, Scott Labs, USA), Lysozyme (0.5g/L, BSG Select Ingredients, Shakopee, MN, USA), and Bactiless (0.5 g/L, Lallemand, Memphis TN, USA) solutions were prepared, mixed, and added to the musts. These were added to prevent contamination and improve maceration. Potassium metabisulfite was used to add 20 ppm of sulfur dioxide ($SO_2$) to the must afterwards. The must was mixed to homogenize the solutions and the fermenters were stored at room temperature overnight.

On yeast inoculation day, the soluble solid content (SSC) in °Brix was tested in the must using an Anton Paar DMA 35 density meter (Anton Paar, Graz, Austria). Corn sugar (Brewmasters, Pittsburg, CA, USA) and drinking water were added and adjusted until the final °Brix reached about 22 and the volume of each fermenter was 3 L. In total, 0.5 kg of blackcurrant was used for 3 L of final must, which represents 16.6% (*w/v*) blackcurrants for each fermenter.

Three treatments were carried out in triplicate using three different yeast strains: (1) *T. delbrueckii*, strain Biodiva, followed by *S. cerevisiae*, strain EC1118, as the sequential inoculation (referred to as Biodiva, hereafter); (2) *M. pulcherrima*, strain Flavia, followed by EC1118 as the sequential inoculation (referred to as Flavia, hereafter), and (3) sole inoculation with EC1118 (referred to as EC). The Biodiva, Flavia and EC1118 strains were purchased from Scott Lab, CA, USA. For each sequential inoculation, EC1118 was inoculated 24 h after Biodiva or Flavia inoculation. Each treatment contained 3 replicates (fermenters). Yeast (0.27 g/L) and Go-Ferm Protect (0.34 g/L) were inoculated after the sugar was homogenized in each fermenter. Sequential inoculation was carried out, when applicable, 48 h after the first strain was inoculated. Fermaid O (Lallemand Inc., Montreal, QC, CAN) as a yeast nutrient supplement was added after 3 days of EC1118 inoculation at the rate of 0.264 g/L, following the manufacturer's protocols.

Wines were racked into new containers twice during fermentation. The first time was when the °Brix decreased to approximately 3. The second time was for transferring

the settled wine as part of clarification. At the second transfer time, when ethanolic fermentation was complete, 50 ppm $SO_2$ was added via potassium metabisulfite to prevent contamination. Afterward, the wines were marked and bottled.

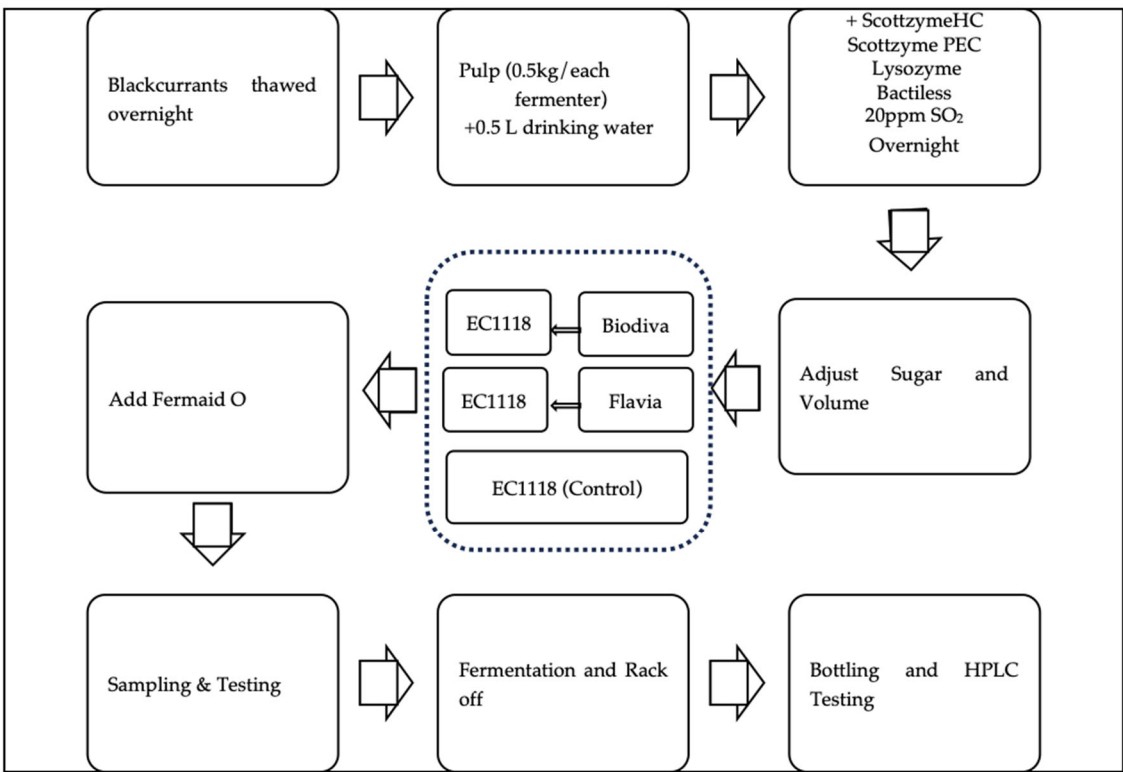

**Figure 1.** Blackcurrant fermentation flow chart with three different yeast inoculation treatments. The three treatments (in dash rectangle) are (1) sequential inoculation of Biodiva and EC1118 stains; (2) sequential inoculation of Flavia and EC1118 stains; (3) inoculation with EC1118.

*2.2. Test Sample Collection*

Samples (15 mL) were collected from each fermenter daily during the first 10 days, on day 17 of fermentation, and when blackcurrant wines were finished. Samples were stored at −20 °C until analysis. For HPLC tests, the final samples were collected and transferred into 236.5 mL Amber Boston rounds with a black poly cone cap (Glass Bottle Outlet, Lake Placid, FL, USA) after purging argon. Other test samples of final wine were also collected into 15 mL plastic centrifuge tubes at the same time and stored at −20 °C until analysis.

*2.3. Sample Analysis*

Must sample sugar content was evaluated by Sucrose/D-Fructose/D-Glucose Assay kits (Neogen Megazyme, Lansing, MI, USA). Blackcurrant must acids were tested using L-Malic Acid Kits and Citric Acid kits (Neogen, Megazyme, MI, USA). Total Yeast Assimilable Nitrogen (YAN) was the combined amount of Primary Amino Nitrogen (PAN) and YAN$_{AUG}$. YAN$_{AUG}$ was tested using an L-LARGE (L-Arginine/UREA/Ammonia) kit from Neogen, Megezyme, MI, USA. A Primary Amino Nitrogen Assay Kit (Neogen, Megezyme, MI, USA) was used to measure PAN in the must. A SPECTROstar Nano microplate reader (BMG Labtech, Ortenberg, DEU) was used to capture the UV spectrum for each test.

Must pH was tested with a PAL-pH digital hand-held pocket pH meter (Atago, Tokyo, JPN). Soluble Solid Content (SSC, °Brix) was obtained using a DMA 35 density meter (Anto Paar, Graz, AUT).

The final blackcurrant wines were analyzed using an FTIR wine analyzer (Lyza5000 wine, Anton Paar, Graz, AUT). The profiles of wine ethanol, glycerol, and pH were collected. Final wine titratable acidity was assessed for a 5 mL sample titrated using 0.1 N sodium hydroxide until the pH reached 8.20 [27]. The results were calculated as grams of citric acid equivalent/100 g.

### 2.4. Anthocyanins and Phenolics Assays on Blackcurrant Musts

Total phenolics were quantified through the Folin Ciocalteu method with a polyphenol quantification assay kit (KB03006, BQC, Asturias, ESP). The outcome data were expressed as $\mu$g Gallic Acid Equivalent (GAE)/mL. Total anthocyanins of blackcurrant must were detected with an Anthocyanins Assay kit (KB03015, BQC, Asturias, ESP). The outcome data of anthocyanins were expressed as mg/L. The must from each fermenter was used for both assays following the manufactures' protocol booklet for each assay kit.

### 2.5. Chromatic Properties

During fermentation, blackcurrant must from each of the nine fermenters was collected into a 1.5 mL centrifuge tube to assess fermentation dynamics. Centrifuge tubes were centrifuged for one minute at the speed of 4000 RPM, 980× *g*, with a centrifuge (MC12-pro, LAB FISH®, Taipei, TWN) before transferring 370‡$\mu$L of supernatant into Corning 3997 96-well plates (Millipore Sigma, Rockville, MD, USA) for assessing the visible color spectrum. The color spectra from 380 nm to 700 nm with 1 nm intervals were analyzed using a SPECTROstar Nano microplate reader. The collected spectra from the microplate reader were converted by the software ColorBySpectra [28]. The absorbance results from the plate reader were translated into CIEL*a*b* color space. Illuminant D65 was the standard used for the data translation. The color data were reported with coordinates of L* (lightness), a* (green-red), and b* (blue-yellow). The dynamic changes during fermentation were plotted with R version 4.2.1 [29].

### 2.6. Antioxidant Capacity of Blackcurrant Wines

The antioxidant activity of blackcurrant wines was evaluated using a 2,2-diphenyl-1-picrylhydrazyl (DPPH) Antioxidant Assay Kit (Colorimetric) (ab289847, Abcam, Cambridge, GBR). Following the manual of the product, the results were represented as Trolox Equivalent Antioxidant Capacity (TEAC eq. $\mu$M/$\mu$L).

### 2.7. Monomeric Phenolic Analysis by HPLC-DAD

Monomeric phenolic compounds of blackcurrant wines were analyzed using a 1260 Infinity II HPLC (Agilent Technologies, Santa Clara, CA, USA) with a reversed-phase column (LiChrospher 100-5 RP18 250 mm × 4.0 mm, 5 $\mu$m, Agilent Technologies, Santa Clara, CA, USA), DAD (Agilent 1260 Infinity II DAD WR) (Agilent 1260 Infinity II FLD Spectra), as previously published [30,31]. The mobile phases were 50 mM ammonium dihydrogen phosphate pH 2.6 (mobile phase A), 20% (*v/v*) mobile phase A in acetonitrile (mobile phase B), and 0.2 M orthophosphoric acid in water, pH 1.5 (mobile phase C). The detailed gradient followed the outline given in the previous publication [32]. The column temperature was maintained at 40 °C with a flow rate of 0.5 mL/min. An amount of 20 $\mu$L of sample supernatant was injected. The monomeric phenolic compounds were identified and quantified at different wavelengths: 280 nm for flavanols, 316 nm for hydroxycinnamic acids, 360 nm for flavonols, and 520 nm for anthocyanins. Flavan-3-ols were quantified using (−)-epicatechin (E1753, Sigma-Aldrich, St. Louis, MO, USA) as the reference standard. Hydroxycinnamic acids were quantified using caffeic acid (C0625, Sigma-Aldrich, USA) as the reference standard. Flavanols were quantified using querce-

tin-3-O-glucoside (E4018, Sigma-Aldrich, USA) as standard. Anthocyanins were quantified using cyanidin-3-O-glucoside as standard.

### 2.8. Statistical Analysis

All experimental data were expressed as mean ± standard errors. Statistical analysis was conducted using R version 4.2.1 [29]. One-way ANOVA was performed for each treatment for the final wine to determine the statistical significance within a 95% confidence level. The least-square mean values were separated using the emmeans 1.8.5 package and letters of significance were identified for pairwise comparisons using the lsmeans 2.30-0 package [33,34].

### 3. Results and Discussions

#### 3.1. Pre-Fermentation Content

Based on the pre-fermentation content of sugars in blackcurrant must after sugar adjustment (Table 1), glucose and fructose were the main fermentable sugars in the fermenters, especially D-glucose, which represented the majority of total sugars. Based on the enzymatic assay results, there was no sucrose detected. There was no detection of sucrose in this study, whereas in previous publications, fructose was the main sugar in blackcurrant cultivars, followed by glucose, and with a small amount of sucrose detected [35,36]. There might be two reasons for the lack of detection of sucrose in this study. HPLC tests were used in the published research, which are more sensitive to detecting trivial amounts of certain sugars. Also, in this study, to produce ameliorated wines, 0.5 kg blackcurrant mixture (juice with pulp) was used in each 3L fermenter, which only contributed about 16.66% of the must weight. In this research, glucose represented most of the fermentable sugars, mainly coming from the addition of corn sugar.

Organic acid quantification in the blackcurrant must indicated that citric acid was the main organic acid, at about 6.57 g/L, followed by malic acid (0.13 g/L). This corresponded to the acid screening in blackcurrant fruits [35].

In previous fermentation research with blackcurrant 100%, the juice pH was only 2.96, slightly outside of normal wine's parameters [37]. In this study, the must from 0.5 kg blackcurrants added to the 3L fermenter had a pH of approximately 3.03, which was in the range (pH 3.00–3.60) required for the most stable fruit fermentation. Therefore, dilution ameliorated the pH and acidity in the fermentation must to acceptable bounds.

The YAN content in the must was much lower than the YAN requirement for *Saccharomyces* strain fermentation and non-*Saccharomyces* fermentation [38]. At least 120–150 mg N/L is recommended for healthy fermentation via most yeast strains. Therefore, for blackcurrant fermentation, the addition of nutrients such as nitrogen is essential for a healthy fermentation.

**Table 1.** Pre-fermentation content in the must after sugar adjustment.

| | **Content** | | |
|---|---|---|---|
| | D-Glucose (g/L) | D-Fructose (g/L) | D-Sucrose (g/L) |
| | 206.10 ± 5.55 | 4.65 ± 0.81 | ND |
| Blackcurrant Pre-fermentation Must | Malic Acid (g/L) | Citric Acid (g/L) | pH |
| | 0.13 ± 0.04 | 6.57 ± 0.18 | 3.03 ± 0.02 |
| | Anthocyanin (mg/L) | Phenolics (mg GAE/mL) | YAN (mg/L) |
| | 6.42 ± 0.34 | 3.88 ± 0.19 | 5.71 ± 1.22 |

Values are listed as mean ± standard error of replicates. Three treatments and three replicates were included. Corn sugar was used for the adjustment of °Brix. The tests were based on enzymatic (Megazyme) activities. ND means not detected by enzymatic methods.

### 3.2. Fermentation Soluble Solid Content (SSC) Dynamic Changes

Overall, Biodiva-treated musts were slower in consuming the sugar (Figure 2). Flavia and EC1118 had quicker fermentation speeds compared to the Biodiva treatment. The differences were larger after one week, as Biodiva required further time to complete fermentation.

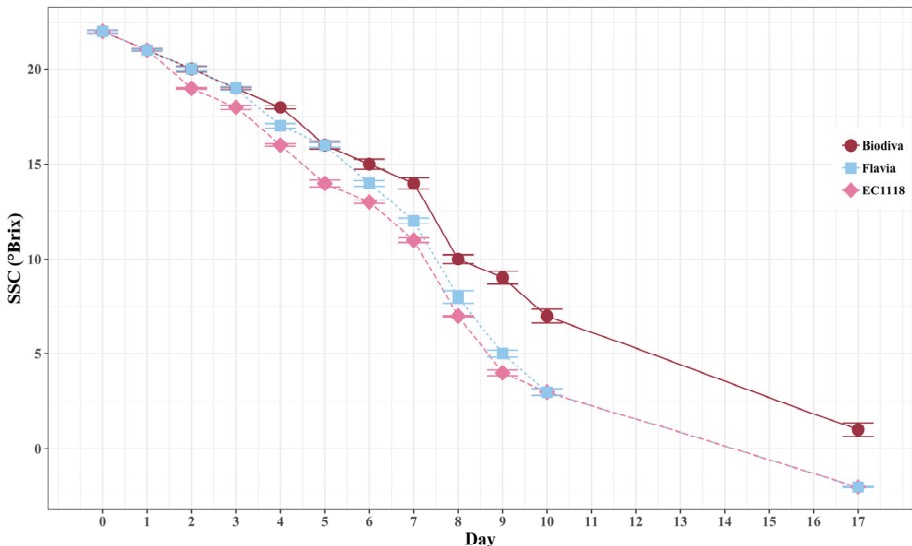

**Figure 2.** Soluble solids (SSC) °Brix dynamic changes during fermentation. There were three treatments, including: (1) Biodiva—Biodiva, EC1118 sequential inoculation; (2) Flavia—Flavia, EC1118 sequential inoculation; (3) EC1118—EC1118 single strain inoculation. Bars indicate the standard error of the mean for in individual treatment and three replicates are included in each fermentation day under each treatment.

### 3.3. Color Dynamic Changes

The colors during fermentation (Figure 3) indicate the treatments altered color dynamics and final wine colors. Throughout fermentation with the sequential inoculation of Biodiva and EC1118, the lightness (L*) of must showed a decreasing trend before day 4, but afterward, Biodiva must showed increased lightness (Figure 3a). This also can be seen in the converted 1 cm color in Figure 3d. Flavia sequential inoculation slightly increased the lightness compared to EC1118 single inoculation across most of the fermentation days. EC1118 showed dramatically decreasing lightness during the first two days of fermentation. From the a* values, we see that Biodiva also showed a different trend compared to the other two treatments, with decreases in a* in later dates (Figure 3b).

Similarly to the L* and a* values, b* values also showed large changes in Biodiva-treated wines. The b* values under the Biodiva treatment consistently decreased after day 4, and the final b* value was lower than in the other two treatments, with only 22 b* values on average. Whereas both EC1118 and Flavia treatments were increased gradually during the first four days, afterward, the b* values only slightly reduced with the fermentation process. At the bottom of Figure 3d, the final wine color after two months of storage indicates all the wines became brighter than they were throughout fermentation.

It has been reported that anthocyanin degradation contributes to the color change during fermentation [39]. In mulberry wine fermentation, the color changed the most in the first two days [40]. In this research, Biodiva showed a different influence on fermentation dynamics compared to the other two treatments. Flavia and EC1118 treatments showed similar color changing trends during fermentation. The underlying reasons for Biodiva's differential performance might be related to its different growth rates and metabolism [41]. Further research needs to be conducted to confirm this hypothesis in blackcurrant wine fermentation.

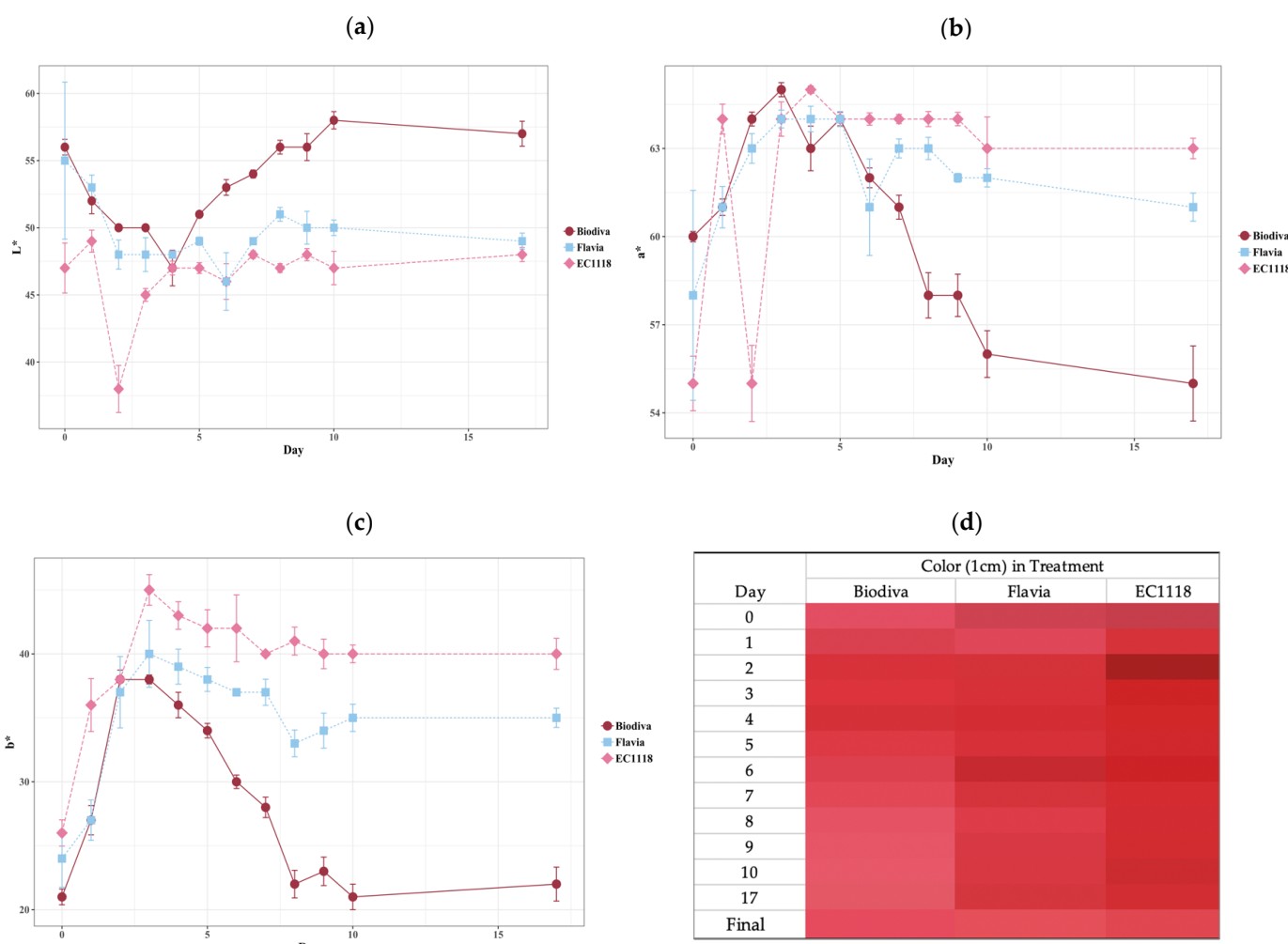

**Figure 3.** Blackcurrant fermentation color parameter changes. Three color parameters are given in the figures, including (**a**): L* (lightness); (**b**): a* (green to red); (**c**): b* (blue to yellow); (**d**): juice color during fermentation (1 cm). The three treatments were (1) Biodiva—Biodiva followed by EC1118 yeast strain inoculation; (2) Flavia—Flavia followed by EC1118 yeast strain inoculation; (3) EC1118—only EC1118 yeast strain is inoculated into the fermenters. Colors changed during fermentation. Bars in the figures indicate the standard error of the mean for individual treatment, n = 3. The color in (**d**) indicates the must average real color in 1 cm cuvettes during fermentation days (Day). The final wine color after two months of storage is listed in (**d**).

*3.4. Final Wine Composition*

The different microbial inoculation treatments caused differences in the final wine's chemical characteristics (Table 2). In previous publications, residual sugars have been a point of focus following Biodiva fermentation, which might be due to the *T. Delbrueckii* yeast strain having fermentation nutrient preferences or preferred sugar types [17,42,43]. From the final wine residual sugar data (Table 2), we see that the Biodiva treatment contained significantly more residual sugar compared to the other treatment, although the residual sugars in all the final wines were less than 1 g/L. The EC1118 single strain inoculation and Flavia sequential incubation treatments had no significant impacts on residual sugar.

Biodiva–EC1118 sequential inoculation gave rise to a lower pH (3.23) than the other two treatments. Meanwhile, Biodiva-treated wine also showed distinct titratable acidity (about 6.91 g/L), higher than the wines from the other two treatments. Biodiva did not give rise to significant differences in final wine ethanol and glycerol contents compared to the Flavia treatment, but it did cause significant differences compared to EC1118 single

inoculation-treated wine. EC1118 single strain inoculation yielded fermented wines with higher ethanol (about 13.39%), but lower glycerol (about 7.23 g/L). The results might indicate the yeast strains had differential fermentation metabolisms [44]. The divergent production between ethanol and glycerol pathways could explain the higher ethanol, but lower glycerol, in the final wine.

It has been reported that Biodiva could increase the glycerol and decrease the ethanol content in the final wines [16,19]. Similarly, this research has also demonstrated Biodiva differentially impacts blackcurrant wine relative to standard *Saccharomyces* spp. fermentation. Meanwhile, it has been reported that Flavia also reduces alcohol and volatile acidity while increasing the glycerol content of wines [16,17]. In summary, Biodiva and Flavia both resulted in slight decreases in ethanol content and increases in glycerol compared to EC1118 alone in blackcurrant wine fermentation.

The final wine pH in this study ranged 3.23–2.27, whereas earlier fruit wine research on 100% blackcurrant juice fermentation yielded a pH of 2.99 caused by *S. cerevisiae*, and of 3.06–3.07 when using non-*Saccharomyces* strain treatments. All were lower than the final wine pH in this study [37]. Meanwhile, the undiluted blackcurrant juice research resulted in a blackcurrant wine with high titratable acidity (36.2 g/L), but the blackcurrant wine from this research only showed 3.23–3.27 g/L acidity, indicating that dilution might produce more approachable and readily consumable blackcurrant wines with *Saccharomyces* and non-*Saccharomyces* strains. The evaluation of analytical parameters, including physical, chemical, microbiological and sensory parameters, is commonly used in testing quality fruit wine [45]. Therefore, sensory evaluation is a worthy component to build into future studies.

**Table 2.** Final wine's basic characteristics.

| Treatment | Total Sugars (g/L) | pH | Titratable Acidity (g/L Citric Acids) | Ethanol (%vol) | Glycerol (g/L) |
|---|---|---|---|---|---|
| Biodiva | 0.93 ± 0.06 [a] | 3.23 ± 0.01 [b] | 6.91 ± 0.13 [a] | 13.14 ± 0.05 [b] | 7.63 ± 0.06 [a] |
| Flavia | 0.60 ± 0.1 [b] | 3.27 ± 0.02 [a] | 6.10 ± 0.15 [b] | 13.23 ± 0.03 [b] | 7.53 ± 0.06 [a] |
| EC1118 | 0.47 ± 0.06 [b] | 3.26 ± 0.01 [a] | 5.89 ± 0.13 [b] | 13.39 ± 0.04 [a] | 7.23 ± 0.15 [b] |

Total sugars, pH, ethanol, and glycerol were detected with an FTIR wine analyzer. Titratable acidity is represented as citric acids (g/L). Three replicates are included in each treatment. Tannin is detected by enzymatic methods. Different letters in the same column indicate significant differences at $\alpha = 0.05$.

### 3.5. Antioxidant Capacity of Final Wines

The antioxidant scavenging DPPH assay results (Figure 4) of blackcurrant wines are similar among treatments, ranging from 105 to 107.2 TEAC eq. μM/μL. In this study, the non-*Saccharomyces* strains and *Saccharomyces* stain did not cause significant differences in the wine's antioxidant capacities.

It was reported that non-*Saccharomyces Hanseniaspora uvarum* and *Starmerella bacillaris* produced Madeira wine that had higher antioxidant potential [46]. It has also been reported that non-*Saccharomyces* yeasts increase glutathione concentration, and glutathione has strong antioxidant activities [47]. Melatonin, a bioactive compound with antioxidant properties, can be synthesized in different amounts using non-*Saccharomyces* and *Saccharomyces* strains [48]. The antioxidant capacity is highly correlated with yeast strains, and the variously significant impacts are different based on detection methods, fermentation materials, and conditions [49]. Further studies are necessary to find the specific bioactive compounds produced by non-*Saccharomyces* and their antioxidant capacities.

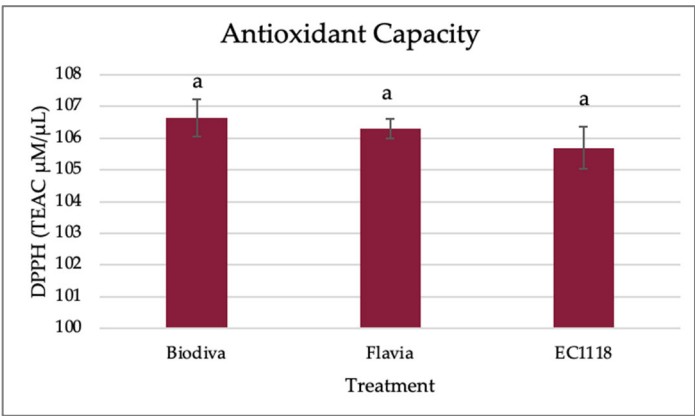

**Figure 4.** Blackcurrant wine antioxidant capacity. The antioxidant capacity (TEAC eq. μM/μL) of final blackcurrant wines was measured by DPPH assays. TEAC represents Trolox Equivalent Antioxidant Capacity. Three replicates were included in each treatment. Values are listed as mean ± standard error. Bars indicate the standard error of the mean for each individual treatment with three replicates. The same letters above the error bars indicate no significant differences at $\alpha$ = 0.05.

*3.6. Monomeric Phenolics of Final Wines*

From the monophenolic profiles (Figure 5), we see that EC1118-fermented wine had the highest flavanol concentration and the lowest concentration of hydroxycinnamic acids and anthocyanins. Biodiva–EC1118 sequential inoculation produced blackcurrant wine containing more total hydroxycinnamates and total anthocyanins. Increases were observed in the total anthocyanins involved in sequential fermentation by *T. delbrueckii* compared to *S. cerevisiae* controls in red wine fermentation [50,51]. For fruit wines, it has been reported that *T. delbrueckii* has enhanced blueberry anthocyanin levels, which result corresponds to the findings of this study [52].

Treatments did not cause significant differences in total flavonols. The flavanol–anthocyanin products have been reported to influence the color and quality of blackcurrant juices [53]. In this study, the differential anthocyanins and flavanol contents might be the main factors influencing blackcurrant wine's color and quality. This might elucidate some of the factors contributing to Biodiva-inoculated final wines with more red pigments (Figure 3).

Flavonoids and their compounds in blackcurrants are of particular importance to their antioxidant activities [54]. Blackcurrant varieties and fruit ripeness largely determine the main flavonoids and antioxidant properties [55]. Here, this study also showed that yeast strain influenced the flavonoid content in blackcurrant wine, especially as regards anthocyanins and flavanols, but there were no significant differences in total flavonols. Nonflavonoids, the total hydroxycinnamates, were also not affected by yeast strains. In terms of wine aroma, both *T. delbrueckii* and *M. pulcherrima* were reported to result in higher concentrations of thiols [56,57]. Further sensory characteristics need to be investigated in the context of blackcurrant wine fermentation.

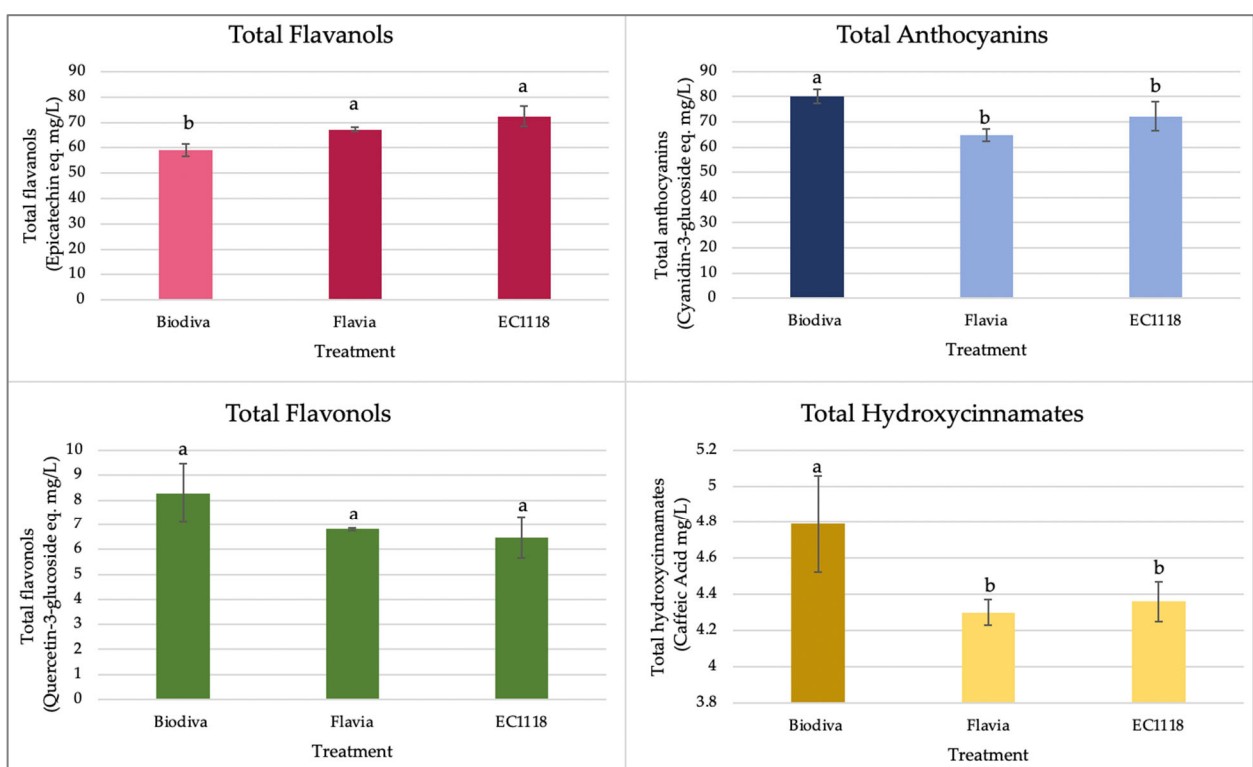

**Figure 5.** Monophenolic profiles of blackcurrant wines. Three treatments are included in each profile: (1) Biodiva—Biodiva, EC1118 sequential inoculation; (2) Flavia—Flavia, EC1118 sequential inoculation; (3) EC1118—EC1118 single strain inoculation. Bars indicate the standard error of the mean for individual treatment with three replicates. The same letters above the error bars indicate no significant differences at $\alpha$ = 0.05. Color hue differences indicate different groups of monophenols. Different colors indicate different monophenol categories. Color lightness differences indicate significant differences.

## 4. Conclusions

In conclusion, the present work is the first study on the influence of non-*Saccharomyces* and *Saccharomyces* on blackcurrant wine quality, including wine color, basic physiochemical characteristics, antioxidant capacity, and monophenolic profiles. The sequential inoculation of Biodiva–EC1118 and Flavia–EC1118 resulted in similar sugar dynamic changes during fermentation as solo inoculation using the EC1118 strain, with Biodiva-treated wines being slightly slower to complete primary ethanol fermentation. Biodiva caused differential color dynamic changes compared to the other two treatments. The final blackcurrant wine produced with Biodiva treatment was brighter, but with fewer red and yellow pigments. On the other hand, Flavia–EC1118 and EC1118 treatments led to similar color dynamic changes during fermentation.

The final wine characteristics indicate that Biodiva caused significant differences in pH and total acidity. Inoculation with EC1118 alone led to a higher ethanol but lower glycerol content in the final wine, which indicates that sequential inoculation caused yeast fermentation process changes. The final wine's antioxidant capacities and monophenol profiles indicate that sequential inoculation could influence total flavonoids but not nonflavonoids.

Blackcurrant's potential health benefits are well known, but its value-added products and wine production methodology remain poorly understood by producers and researchers, especially as regards the formulation and formation of blackcurrant wines. Our findings provide useful information elucidating *Saccharomyces* and non-*Saccharomyces* applications in blackcurrant wine fermentation.

**Author Contributions:** Conceptualization, Z.W.; methodology, Z.W. and A.S.; investigation, Z.W., A.S. and A.A.W.; formal analysis, Z.W. and A.S.; resources and fruit collection, Z.M. and A.S.; writing—original draft, Z.W.; writing—review and editing, Z.W., A.S., Z.M. and A.A.W.; funding acquisition, Z.W., A.S. and Z.M. All authors have read and agreed to the published version of the manuscript.

**Funding:** This research is funded by the Montana Department of Agriculture. Grant No.: 23SC00309.

**Institutional Review Board Statement:** Not applicable.

**Informed Consent Statement:** Not applicable.

**Data Availability Statement:** The datasets generated and/or analyzed during the current study are available from the corresponding author on reasonable request.

**Acknowledgments:** The authors would like to thank Farm Manager Haydon Davis and the 2022 summer crew for assistance with farm maintenance. The authors would also like to thank Research Associate Bridgid Jarrett and Administrative Associate Kierstin Schmitt for the work coordination.

**Conflicts of Interest:** The authors declare no conflicts of interest.

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
