# Peer review of "Assessment of Sequential Yeast Inoculation for Blackcurrant Wine Fermentation"

_fermentation, doi:10.3390/fermentation10040184_

Round 1

Reviewer 1 Report

Comments and Suggestions for Authors In this paper, comprehensive studies were conducted after non-Saccharomyces and Saccharomyces yeast strain inoculation in less than 20% fruit must for blackcurrant fermentation, which provided useful information for the yeast strain application in blackcurrant wine fermentation. And, it has important guiding significance for the deep processing of blackcurrant. The design of the paper is reasonable, the data are detailed, and the results can support the conclusion of the paper. I think it is acceptable for publication after minor revisions as follows: 1. In Figure 2, error lines should be added, and the horizontal scale should be more intuitive. 2. There are too many references, and it is recommended to keep 30-40 of the most critical documents. 3. Please try to distinguish the results of different experimental groups with different colors in Figure 3 and Figure 4.

Reviewer 2 Report

Comments and Suggestions for Authors

The authors present a study describing the effects of sequential inoculation of non-Saccharomyces and Saccharomyces yeasts on blackcurrant wines. The use of these winemaking techniques is of general interest, however, parts of the paper are poorly written and show serious flaws. It needs to be revised substantially before it should be considered for publication.

General comments

The introduction is going back and forth between present tense and past tense. It needs to be consistent following a more scientific writing style overall. There is also a lot of redundant information throughout the paper. Some sentences appear almost identical in different parts of the manuscript, sometimes only a few sentences apart. It almost seems like the authors did not proofread the manuscript as a whole before submitting it. 

Please refer to the guide for authors on how to reference materials in your method description. It needs to be consistent and descriptive, as well as mention the manufacturer and supplier accurately including city, state an country.

Some tables and figures are very blurry and incorrectly formatted. Please improve the quality of all figures and tables and refer to the manuscript template for formatting questions.

Specific comments

Line 24  How can yeast stimulate anthocyanin and hydroxycinnamate concentrations? I do not think that stimulate is the right word here.

Line 35  Should be vibrant.

Line 48  This whole part is redundant.

Line 53  Why is the different anthocyanin spectrum important? It should at least be mentioned here if it is relevant.

Line 66  Copigmentation is very controversial in the wine industry. Even the original photometric assay is producing artifacts that are irrelevant for actual wine color.

Line 74  After yeast was already discussed above, this paragraph is going back to it. The whole introduction is unorganized and needs to be revised and rewritten.

Line 83  What is glycerol et al.?

Line 86  This was just mentioned above and is redundant again.

Line 88  Volatile compounds are all flavor related. This statement does not make sense.

Line 89  As far as I know, all of these yeast trade names are actually trademarked and need to be referenced as such. The same is true throughout the manuscript.

Line 103 "No research has been evaluating" is a pretty absolute statement. Add "to the best of our knowledge" to it or rephrase the sentence.

Line 107 The majority of this paragraph should be in methods and materials, not in the introduction.

Line 116 The objective of the study is still not entirely clear and should be stated clearly.

Line 122 Section 2.1 is not a design description and should be deleted.

Line 127 The whole section 2.2 is written poorly and is not descriptive enough to fully understand what the authors did here. It needs to be revised completely.

Line 183 Figure 1 is completely redundant because all information is already somewhere in the text.

Line 188 Section 2.3 is entirely redundant. All the information is already listed at the end of section 2.2.

Line 197 Which instrument was used with all these kits listed under 2.4?

Line 208 The standard endpoint for weak acids is 8.2

Line 216 The last sentence of 2.5 does not make sense. What is the original must? What instructions were followed?

Line 267 Please do not start the section with the table. Also, the table needs to be reformatted to match the style guidelines of the journal. Why is the standard deviation for citric acid so high?

Line 284 The phrase "took the most part" appears multiple times in this section. This is not good scientific writing. I understand what the authors are trying to say but it is incredibly hard to read.

Line 286 The whole paragraph about acidity and pH needs to be revised. It is confusing and unorganized.

Line 296 This is completely outdated information. The nutrient requirements for yeast are currently referenced as 10ppm of N per degree Brix. The 150ppm threshold was abandoned about 20 years ago.

Line 309 The Figure is incredibly hard to see and should be revised. There also need to be more discussion and statistics. Are those differences actually significant?

Line 333 Please add statistics to all the figures. Figures should be listed as 3a, b, c and d.

Line 344 I do not see where this information comes from because the quality of Figure 3d is horrible. Also, please be careful with the term pigment here since you have not analyzed pigments.

Line 352 The value did not slide down. Please revise the language in this entire paragraph.

Line 365 That statement is not true. There are plenty of studies out there that describe color dynamics during wine fermentation, especially the color loss towards the end.

Line 373 This whole paragraph needs to be rewritten. Where is the connection between growth rate and color?

Line 377 Do not start a section with a table.

Line 401 You did not analyze amino acids or esters. How does the literature support your results?

Line 404 I am not sure that I would state this as proving differences in metabolism. Please revise these absolute statements throughout the discussion. Proving something takes a lot more repetitions and statistical work.

Line 423 The correlation between polyphenol concentration and antioxidant capacity is poor at best. Please discuss the data differently here.

Line 469 A comprehensive study would have to include sensory evaluation. This is, again, to absolute to be true.

Line 476 Without statistics, it is hard to say which of these statements is actually significant.

Line 484 You did not study any health benefits and your data has gaps, so this should not be your final conclusion.

Comments on the Quality of English Language

The language is very hard to follow at times, especially in the methods and materials section and parts of the results. It is essential that the manuscript is edited by a native speaker before it should be reevaluated for its scientific soundness since some parts are simply not written well enough to be comprehensible. 

Reviewer 3 Report

Comments and Suggestions for Authors

The article by Wang et al. provides a baseline analysis of blackcurrant wine fermentation with non-Saccharomyces and Saccharomyces yeast strains. The data indicate Biodiva had a significant impact on wine pH and acidity, and although total antioxidants were similar among all groups, Biodiva increased total anthocyanin and hydroxycinnamates content and  reduced total flavanols. These results further support the generally accepted and understood fact that fermentation outcomes differ with yeast strains. The article could be improved by editing for conciseness and english grammar, including word selection. 

Comments on the Quality of English Language

The article could be improved by editing for conciseness and english grammar, including word selection. 

Reviewer 4 Report

Comments and Suggestions for Authors

Dear authors,

The manuscript “Assessment of Sequential Yeast Inoculation for Blackcurrant” is generally well written and deals with a very interesting topic, namely the use of non-Saccharomyces yeasts in fruit wine fermentation and how they can influence the final product. While I find the topic very interesting, there are a number of points that need to be improved (and even added to) before it can be considered for publication.

The most important point is that, in my opinion, it is necessary to include sensory analysis of the products obtained. In fact, the authors have made different remarks in this regard (e.g. lines 416-417, 466-467) but I consider that it is necessary not only to indicate it as a future work, but to make at list a first approximation (e.g. triangular test). Given that the product obtained must be accepted by consumers, and that it is even part of the hypothesis (as described in the introduction section), I consider that this manuscript is incomplete without the sensory part.

In Material and Methods section, I see some weaknesses. The first subsection, Experimental design, instead of making the sense of the work clearer, makes it difficult to understand what the experimental design consists of, which is in fact graphically explained in Figure 1 (it should be considered to move this figure to the subsection instead). Please consider modifying it (by adding information found in 2.2 and therefore removing it from this sub-section) or deleting it. If amended, please note that the word “strain” should not be italicised.

When inoculating with yeast strains, what concentration of each yeast was inoculated? I don't understand the sentence on line 147, which was inoculated when the sugar content became stable? It seems to me to be an imprecise term.

Was any microbiological monitoring done? Was the co-inoculation with Saccharomyces yeast done in all cases at 24 hours, regardless of how the fermentation started, or the viability of the yeasts? Perhaps the non-Saccharomyces yeasts did not even have time to implant before inoculation with the Saccharomyces yeast, which in turn would explain the few significant differences found in the analyses performed.

Thus, considering the suggested changes, the abstract should be also modified to better focus the present study.

Other concerns:

L83: There is a mistake, “glycerol et al.”

L88-89: This sentence is too short for me and is not really contributing much. What potential variable? Did they use different ratios of Saccharomyces and non-Saccharomyces yeast in the co-inoculation? If so, perhaps this paper is more interesting to use to discuss the results of this manuscript than the introduction section (which in my opinion is complete and focused as it stands).

L94-97: Please check these sentences, I think there is a grammatical problem that makes it hard to fully understand the meaning. Bibliographic reference?

L110-112: Any references?

L163: I’m not sure if you can use “oz”, maybe you should change to SI.

L196-204: Which spectrophotometer was used?

L212-217: Asturias is a province of Spain, so "Asturias, Spain" is the correct term.

L231: The R software quote, number 40, is missing.

L284, L295-299:: Why was only about 17% weight of blackcurrant added to the fermenters?

L362: Sentence cut off: “Followed by Flavia treatment with 35b* values.”

L380: I think there is a mistake: I think there is a mistake: different letters in the same column indicate significant differences, right?

L386: Sentence cut off: “From the final wine residue sugar data (Table 2).”

L389: Co-culture treatment” or co-fermentation, not coincubation.

L403: base wine? Please explain this, I think it is very interesting but needs to be put in context (and explain why it can have such positive effects).

L412: Which previous research?

L421: It should be included, along with the mean, the sd and the units.

L427: I think it would be interesting to further discuss why the antioxidant capacity of the final products was not influenced by strain, if previous studies have described it as strain-dependent.

L447-452: Perhaps it might be interesting to modify the way this discussion is expressed. There seem to be previous studies that indicate different (and contradictory) things, some supporting the results shown in this manuscript and others appearing to differ. If anthocyanins depend on the grape variety, but this does not occur with blueberry, why could this be? Are there more studies on this issue?

Addressing these concerns will strengthen the quality and impact of your work, ensuring its resonance within the scientific community.

Round 2

Reviewer 2 Report

Comments and Suggestions for Authors

The authors did substantial revisions to the manuscript and incorporated almost all of my suggestions. The quality of the work is sufficient for publication, however, the English editing must still be completed by a native speaker.

The only thing that I am still worried about is the use of trade names like Flavia and Biodiva without the TM sign. Please make sure that all trademarked tools and supplies are properly indicated.

Comments on the Quality of English Language

Please let a native speaker perform a language review including grammar and sentence structure.

Reviewer 4 Report

Comments and Suggestions for Authors

Dear authors: 

Congratulations on the changes made. As the current manuscript stands, I think it is more understandable and highlights the findings found of your research.